# Factors affecting clothing purchase intention in mobile short video app: Mediation of perceived value and immersion experience

**Tian Hewei** [ID]*

Department of Fashion, Fuzhou University, Xiamen, China

* thw1949@fzu.edu.cn

## Abstract

Based on the stimulus-organism-response (SOR) framework, this research introduces perceived value and immersive experience, and builds a model of media interaction affecting consumers' consumption of clothing in mobile short video app (MSVA). Among the conducted survey, using the method of questionnaire survey, a total of 820 questionnaires were collected, and data from 752 valid questionnaires were used for analysis. The research results showed that the MSVA media interaction has a positive impact on perceived value, immersion experience, and purchase intention; Perceived value has a significant positive impact on immersion experience and purchase intention; Immersion experience has a significant positive impact on purchase intention. Perceived value and immersion experience play a mediating role in the relationship between social media interactivity and purchase intention. This research will provide theoretical support for clothing marketing businesses of MSVA and suggestions for the development and design of MSVA.

**Data Availability Statement:** All relevant data are within the paper and its Supporting Information files. Raw data is available from the author (thw1949@163.com).

## Introduction

Social media has become an important marketing medium for attracting and retaining consumers, and it is evolving into a social e-commerce platform (Liao & Huang, 2021) [1]. As a new concept of e-commerce, social e-commerce uses social media as a new popular online shopping platform. Users can now view, add to shopping carts, and purchase products in a single social media application (Ahmed, 2021) [2]. Social e-commerce is a relatively new concept that emphasizes e-commerce transactions promoted through social media (Mamonov & Benbunan, 2017) [3]. Now, social e-commerce is an important trend in practice, and it has begun to develop rapidly to provide services to shoppers (Peng et al., 2019) [4]. Its influence among different users around the world is increasing day by day, making them a tool for advertising and e-commerce (Jahng et al. 2007) [5]. In China, mobile short video app (MSVA) widely developed in recent years, such as Tiktok, has become an important marketing channel for many businesses.

The MSVA is a new platform integrating the advantages of social e-commerce and mobile e-commerce. The core innovation of social e-commerce is to use interaction to stimulate

**Funding:** The authors received no specific funding for this work.

**Competing interests:** The authors have declared that no competing interests exist.

consumers' shopping (Miao et al., 2019) [6]. Different from traditional e-commerce, the consumer demand of social e-commerce is usually passive and easy to be induced to consume (Guercini et al., 2018) [7]. Traditional e-commerce searches for product information, and then forms purchase behavior according to consumer needs (Yang et al., 2016) [8]. Businesses usually start with emotional communication, establish a closer relationship with consumers, and then gradually penetrate product information (Kozielski et al., 2017) [9]. Continuous and frequent interaction is the key to the survival and development of social e-commerce (Campbell et al., 2014) [10]. Mobile e-commerce enables consumers to search and buy goods anytime and anywhere (Einav et al., 2014) [11]. The MSVA not only meet the social intercourse needs of consumers, but also meet the mobile shopping needs, therefore, it has gradually become an important marketing position.

Previous studies have explored the influencing factors of purchase intention in social e-commerce (Chen et al., 2018; Hajli, 2019) [12] and mobile e-commerce (Liu & Li, 2019; Chi, 2018) [13]. However, there is no research on the purchase intention of MSVA. In fact, there are many successful cases of MSVA marketing in China. According to Hu (2020) [14], Tiktok is an outstanding representative of MSVA with a great commercial value. As the supply side, short video applications continuously improve user experience by meeting user needs, and finally realize user flow to form a complete closed loop. In the future, short video applications can give better play to business value by stimulating users' potential demand through forward-looking value proposition. MSVA is very innovative in terms of profit creation and marketing, not sold directly, nor will it push a large number of product advertisements, as it realizes commercial purposes through small videos that meet the interests of different users, seeks profitability and economic balance without charging users, and produces greater commercial value in the future (Mhalla et al., 2020) [15]. On the mobile short video sharing platform, MSVA is the best strategy to promote products through user generated short videos. Businesses can attract target customers by publishing a variety of interesting short videos. At the same time, they can also fully interact with consumers in the form of live broadcast to increase product sales (Ge et al., 2021) [16].

With the development of e-marketing, more and more clothing sellers are turning to e-commerce market (Guercini et al., 2018) [7]. Compared with the marketing of other products, clothing marketing needs to highlight the wearing effect. The display of fitting models can fully arouse consumers' shopping intention. In social e-commerce and mobile e-commerce, fashion marketing has accumulated market experience. The rise of MSVA has opened a new direction for clothing e-commerce. The clothes of characters in the video often attract the attention of viewers. Many viewers will leave a message to ask for the purchase link of clothes to realize the clothing marketing of MSVA. In the field of marketing, the specialized research on clothing e-commerce is not particularly sufficient, and the research on clothing marketing of MSVA does not appear.

In order to fill this gap, this research attempts to use the stimulus organization response (SOR) model to explore the influencing factors of clothing purchase intention of MSVA and what the potential reason for this impact mechanism is. This study will provide literature value and practical experience for the research of clothing purchase intention of MSVA. Compared with previous studies, this will focus on the interactivity of MSVA, perceived value and immersion experience, and clothing purchase intention. MSVA include the main characteristics of social e-commerce, mobile e-commerce, and media entertainment. It takes the media characteristics (interactivity) as an external stimulus to infer the generation mechanism of clothing purchase intention of MSVA. At the same time, this study also verifies the scalability of the existing research in an emerging e-commerce model, and lays a theoretical foundation for the follow-up MSVA marketing. At the practical level, the survey results can provide reference

value for developers of new MSVA and provide marketing strategy reference for clothing sellers of MSVA.

## Literature review

**Social e-commerce and MSVA.** Social e-commerce is a new form of e-commerce. With the development of mobile information technology, social e-commerce has become the mainstream forms of e-commerce. At present, scholars have done a lot of research on social e-commerce (Huang & Benyoucef, 2013 [17]; Einav et al., 2014 [11]), but few take clothing e-commerce as an example. Morra et al. (2018) [18] studied the impact of social media marketing on brand equity and fashion brand purchase intention, taking user generated content and company created content on social media as variables. The results showed that user generated content plays moderating a effect in the relationship between company created content and purchase intention. This research is aimed at text social platforms such as micro-blog and Facebook. It is unknown whether the research results are applicable to the current emerging MSVA.

Social e-commerce has attracted the attention of scholars in the field of clothing marketing research. The existing research identified the attraction of clothing social e-commerce to consumers (Nadeem et al., 2015) [19], for Italian Y Generation consumers, and studied how peer recommendation and website service quality affect consumption behavior during Facebook online shopping. The results showed that website service quality affects trust, peer recommendation affects attitude, and the impact on female consumers is significantly greater than that on male consumers. According to Kang and Johnson (2013) [20], social e-commerce realizes clothing social e-shopping through social network stores, by constructing a clothing retail model, discussed the relationship between consumers' self-confidence, loyalty, perceived credibility, and purchase intention. Their research results showed that consumers' self-confidence has a positive impact on the perceived trustworthiness of social e-commerce, perceived trustworthiness has a positive impact on purchase intention, and loyalty plays a regulatory role between perceived trustworthiness and purchase intention.

There is little research on the impact of media interaction of MSVA on clothing purchase intention. In the field of social e-commerce, the current research pays little attention to MSVA. Nash (2019) [21] used qualitative design methods such as in-depth interviews and focus groups to explore the impact of social media on the purchase decision-making process in the clothing retail environment. The results showed that social media has a greater and greater impact on consumers' decision-making. Hasena and Sakapurnama (2021) [22] determined the impact of tiktok's word of mouth on purchase intention of cosmetics through brand image. Changhan et al. (2021) [23] identified the influencing factors of residents' purchase intention through the short video platform Tiktok, and determined that the perceived media richness was the most important factor affecting residents' shopping tiktok short video application. Different from these studies, this paper focuses on the influencing factors of purchase intention of clothing MSVA. The variables of the research framework include media interaction, perceived value, immersion experience, and purchase intention.

**SOR model.** The SOR model was proposed by the environmental psychologists Mehrabian and Russell (1974) [24], where stimuli (S) is the external environmental factor of the organism, organism (O) is a psychological transformation mechanism by which the user internalizes the stimulation into information, and response (R) represents the user to the external stimulus information content of the relevant response behavior. According to Eroglu et al. (2003) [25], these three are the basic elements of the SOR model.

Nam et al. (2021) [26] conducted a cross-cultural study on the impact of online service quality on consumers' trust intention behavior in online clothing shopping. The results

showed that consumers' responsiveness to electronic services under the influence of different cultures is closely related to consumers' trust, and trust is the main factor affecting purchase intention. The research shows that in SOR model, trust is the key intermediary connecting website design, responsiveness, and purchase intention. Gabriella et al. (2021) [27] used SOR model to verify the impact of eWOM and consumer engagement on clothing consumption behavior. The research results showed that eWOM and consumer engagement have a significant positive impact on consumption behavior. In the impact relationship between eWOM and consumption behavior, consumer engagement does not play an intermediary effect. These studies considered online service quality and eWOM as stimuli to explore the impact on purchase intention or purchase behavior, while media interaction act as a stimulus to explore the impact on clothing purchase intention of MSVA.

The SOR model provides a structured research perspective and a solid theoretical foundation for the study of the influence mechanism of consumers on the continuous purchase intention of fashion products on social e-commerce platforms. Based on the rapid development of MSVA business, this study attempts to determine the influencing factors of MSVA consumers' purchase behavior through SOR theory. Considering that the most prominent feature of MSVA is its media interaction, which covers the comprehensive characteristics of application platform system, social e-commerce and mobile e-commerce, the perceived value and immersion are some reflections of individual cognition and emotion. Therefore, this study uses social media interaction of MSVA as the external stimulus, the perceived value and immersion as the organism, and the purchase intention of fashion products as a response.

**Media interactivity.**   Marketers are increasingly using social media platforms as promotion channels. In this way, they prefer highly interactive social media because it allows consumers to better socialize and build networks. According to Lin (2013) [28], based on selected 102 male college students as subjects, media interactivity can significantly affect the behavior of the audience. Through structural equation analysis, it was confirmed that media interaction in video games can significantly improve players' responsiveness; On the other hand, we can know the important value of interactivity. It can enhance the relationship between consumers and machines, consumers and consumers, and consumers and enterprises, induce purchase intention, and realize consumption behavior in the process of interaction.

In the research on e-commerce, many studies discussed interactivity. For example, Li et al. (2020) [29] explored the impact of interactive experience and interactive characteristics on online reputation of e-commerce. Through online surveys, they obtained 345 valid data and investigated five interactive characteristics of consumers to consumers (reciprocity, sociality) and enterprises to consumers (responsiveness, personalization and perceived control), showing that reciprocity, responsiveness and perceived control have a significant impact on online reputation. In the research of Syuhada and Gambett (2013) [30], they believed that the social media used in the commercial form of Facebook in Indonesia is the basis of market interaction, and interactivity can accelerate the word-of-mouth marketing of social networks. According to Chong et al. (2018) [31], the swift relationship created by the interactivity and existence of social media enhances trust and further increases repurchase intention. In the related research of social media, many scholars also emphasized interactivity. From the study of Meng and Leung (2021) [32], 526 sets of data were collected to investigate the role of satisfaction seeking, narcissism, and personality traits in tiktok's participation behavior (i.e. contribution, promotion and creation) in China. The results show that interactivity of social media can enhance users' sense of participation and make users behave more actively. According to Vaterlaus and Winter (2021) [33], taking Tiktok as an example, the influence of interaction on consumers' social media Tiktok stimulates users' pursuit of diversity and game satisfaction.

When carrying out clothing marketing through MSVA, the host or model in short video interacts with consumers, and consumers also have open interaction. This multi-dimensional interaction has a very significant stimulation on consumers. This study aims to explore interaction as a stimulus, rather than denying the existence of other stimuli.

**Perceived value and immersion experience.** There are many research on perceived value in the marketing discipline, mainly focusing on the perceived value of consumers for various products or services. In the fashion marketing discipline, the research on perceived value is not sufficient. Early studies focused on the perceived value of luxury fashion brands (Li et al., 2012) [34], or the difference analysis of the perceived value between luxury and mass fashion brands (Lloyd & Luk, 2010) [35]. In recent years, the research on perceived value in fashion marketing discipline focuses on the impact of self-concept and perceived value on sustainable fashion (Jeong & Ko, 2021) [36], and the impact of brand trust and perceived value on customer satisfaction and purchase intention (Cuong, 2020) [37]. According to Chae (2016) [38], the consumption orientation of fashion products in mobile shopping centers is divided into four categories: convenience / economy, show off / trend, enjoyment, and impulse. Her research revealed that except impulse, other fashion shopping orientations have an impact on perceived value. Different from these studies, we believe that the influencing factors of perceived value are multi-dimensional. When consumers buy clothes in MSVA, media interaction performance improves consumers' perceived value of products. Therefore, we focus on the impact of media interaction on perceived value.

Immersion experience is a concept that appears with the development of internet technology and virtual reality technology. Generally, it refers to that users completely immersed in a certain field to achieve selflessness. For example, users often forget time when playing video games (Jennett, 2008) [39]. This concept is similar to the shopaholic in fashion consumption. Immersion experience has been applied in many disciplines. In the field of education, the impact of immersive experience on learning effect is discussed (Cheng, 2015) [40], and the tourism industry studies the feasibility of the application of immersive experience in virtual travel (Shih, 2015) [41]. Hamilton et al. (2016) [42] investigated the impact of social media brand and consumer interaction on customer value and they believed that consumers' interactive satisfaction and interactive immersion with social media brand can create customer value. Wang et al. (2021) [43] investigated the impact of live broadcast characteristics on consumers' sense of social existence and immersion experience, and believed that the host's charm, interactivity, and trust in the host fully affect the immersion experience. These studies have focused on the positive impact of interactivity on immersion experience, but they have not realized the importance of media interactivity, especially the impact of interactivity of MSVA on perceived value, immersion experience, and purchase intention.

**Purchase intention.** There are many studies on purchase intention of social e-commerce, most of which focus on the influencing factors of purchase intention of social e-commerce. Zhu et al. (2019) [44] used the effect hierarchy model and commitment participation theory, a three-stage model established to evaluate the impact of product cognition on purchase intention. They believed that involvement, situation, and trust all have a positive impact on purchase intention. Chen et al. (2018) [12] identified the influencing factors of purchase intention of social e-commerce through an empirical study. They believed that perceived value and social awareness factors will affect consumers' decision-making and behavioral intention. These studies showed that the research on consumers' purchase intention and purchase behavior has strong theoretical and practical value. Therefore, this study takes purchase intention as response to explore the impact of perceived value on purchase intention.

In the field of clothing marketing, Kamal et al. (2013) [45] investigated whether materialism, an important structure of consumer behavior, is the result of social media use, and also

investigated the relationship between materialism and luxury purchase intention of American and Arab users. The results showed that Arab social media users show a higher level of materialism and social media utilization than American users. Materialism has a positive impact on luxury purchase intention. Escobar-Rodríguez and Bonsón-Fernández (2017) [46] believed that the key factors affecting online fashion purchase intention are perceived value, trust and fashion innovation, and time-saving and perceived security are the main antecedents for predicting perceived value and trust, respectively. These studies on fashion social e-commerce were aware of the impact of social media, perceived value, and other factors on purchase intention, but they did not conduct in-depth research on the characteristics of emerging social media, while this paper studies the MSVA, a new force of social e-commerce and mobile e-commerce.

**Research model and hypotheses.** This research constructs an SOR model of the influence of purchase intention on fashion products in MSVA. As shown in Fig 1, media interactivity is used as a pre-variable, perceived value and immersion experience are used as intermediary variables, and purchase intention is used as a dependent variable.

**Media interactivity.** Social media is highly interactive, breaking the one-way acceptance model of consumers in the traditional media environment. According to the interactive content, it is divided into social interaction and task-based interaction, and according to the utility target of the interaction, it is divided into information-based interaction, entertainment-based interaction, and reward-based interaction. Interactivity has been mentioned many times in the research of information systems, and it is of great significance to the success of communication, marketing, advertising, and commerce. Eggert (2002) [47] verified the impact of interactive experience on user behavior. In view of the characteristics of social media, previous studies have refined interactions into controllability, responsiveness, communication, associativity, and personalization.

Media interaction can enhance consumers' sense of participation through multi-dimensional senses such as vision and hearing. Dong et al. (2018) [48] believed that the interactive information stimulation of online media brings stronger credibility and positive response to consumers, thus affecting brand attitude and purchase intention. Abdullah et al. (2016) [49]

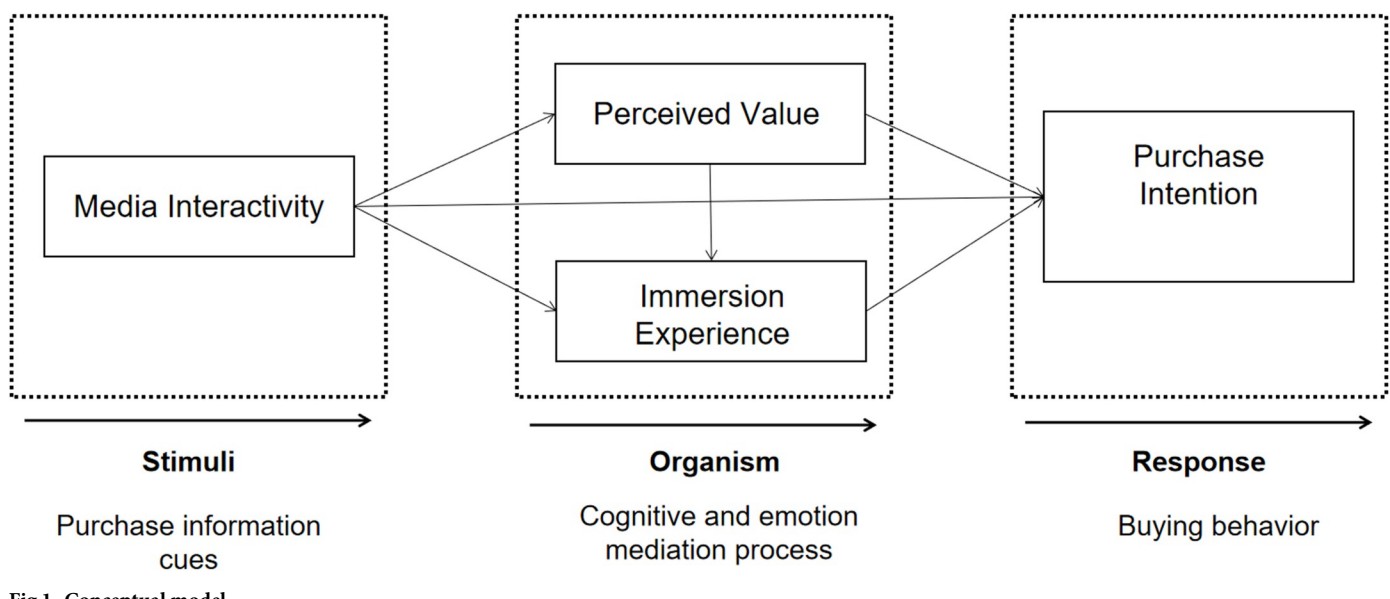

**Fig 1. Conceptual model.**

built a conceptual model for hotel online booking websites, and confirmed that the interactivity of online websites can improve consumers' perceived value and enhance consumers' willingness to revisit, and perceived value plays an intermediary role in the relationship between online website interactivity and revisit intention. Huang and Liao (2017) [50] used the virtual liminoid theory to explore the factors that induce immersion experience in e-shopping environment. The research results showed that the sense of engagement, self positioning, and tactile intention generated by exploring interactive technology can induce consumers' immersion experience. Therefore, the following hypotheses are proposed.

H1: In the MSVA commerce, media interactivity is positively related to fashion purchase intention.

H2: In the MSVA commerce, media interactivity is positively related to perceived value.

H3: In the MSVA commerce, media interactivity is positively related to immersion experience.

**Perceived value.**   Perceived value is the result of consumers' ability to consider products based on the overall perception of gains and losses and product utility evaluation. Perceived value process is the process in which consumers make psychological judgments about costs or benefits based on price comparison in the process of purchasing products or services. From a marketing perspective, perceived value is one of the most effective ways to improve customer satisfaction and maintain continuous purchases.

According to Gan and Wang (2017) [51], in the social e-commerce environment, perceived value has a significant impact on satisfaction and purchase intention, among which utility value has the greatest impact on purchase intention and hedonic value has the greatest impact on satisfaction. Charfi (2014) [52] believed that more and more websites encourage consumers to produce immersion experience, while virtual reality websites can produce immersion experience. Immersion experience is closely related to consumers' hedonic value and practical value, and the sense of participation and professional knowledge play a regulatory role in the relationship between immersion experience and perceived value. Therefore, the following hypotheses are proposed. According to Abdullah et al. (2016) [49], perceived value plays an intermediary role in the relationship between online website interactivity and revisit intention.

H4: In the MSVA commerce, perceived value is positively related to fashion purchase intention.

H5: In the MSVA commerce, perceived value is positively related to immersion experience.

**Immersion experience.**   Csikszentmihalyi (1978) [53] first proposed the concept of flow, which belongs to the research category of positive psychology and has been widely used in other disciplines. Tuncer (2021) [54] opined that flow experience refers to the experience of being completely immersed in an activity, which will produce a sense of pleasure, that is, the overall state of consumers when they are completely immersed in a certain behavior or activity while ignoring other things. In the research of communication, marketing and other related disciplines, mobile experience is mainly called immersive experience. From the perspective of human-computer interaction, the flow experience is the perception of the user during the interaction with the computer. Attention will be completely focused on the interaction, curiosity will be fully mobilized, and the interaction process will be very interesting. The main feature of immersive experience is fun, which is an emotional experience that consumers consider interesting and entertaining.

Lee & Hong (2006) [55] believed that the higher the interactivity, the more it can trigger the immersion experience of online shopping. The higher the immersion experience, the higher the perceived utilitarian value, and the higher the purchase intention of fashion products. Wang et al. (2021) [43] investigated the impact of live broadcast characteristics on consumers' purchase intention in live e-commerce scenarios. The research results showed that the host's charm and trust in the host significantly affect the immersion experience, and the immersion experience significantly affects the consumption intention. Zhou (2020) [56] confirmed that in the social e-commerce environment, human-computer interaction and social interaction have a significant impact on the immersion experience, and then affect the willingness of social purchase and social sharing.

H6: In the MSVA commerce, immersion experience is positively related to fashion purchase intention.

## Materials and methods

### Data collection and sample

This study has been approved by the academic ethics and ethics committee of Fuzhou University (2021120601). I used the survey method to collect data, and the questionnaire survey was conducted in China. In order to obtain scientific and reasonable data, I first conducted a pilot survey, recruited 20 college student volunteers to fill in the questionnaire in Fuzhou University, explained in the title of the questionnaire that the data collection is only used for academic research, and obtained the consent of the participants. The questionnaire content, survey steps and data collection and analysis process met the requirements of the Chinese ethical institutions and academic supervision institutions.

I asked the respondents to answer the questions anonymously. Before answering the questionnaire, there will be an explanation. This questionnaire is only used for academic research. Answering the questionnaire is regarded as agreeing to the academic purpose of collecting data. Our questionnaire does not include minors and does not need the consent of the ethics committee. It is worth noting that according to the preliminary study, the time required to complete the questionnaire is usually about three minutes. The formal survey was published on the questionnaire star platform. The links of the questionnaire are published in short video communication forums, communities, etc. The questions in the questionnaire do not violate the code of ethics, do not involve prejudice, and do not include privacy issues such as subjects' names and contact information. In the questionnaire, clothing products are used as an example of fashion products because the public is very familiar with clothing purchase. In order to ensure the representativeness of the sample results, before answering the questionnaire, the participants were asked whether they had the experience of buying clothes in MSVA. If they have the experience, they will continue to answer the questions. If they have no experience, they will not answer.

A total of 820 questionnaires were received. After excluding the questionnaires with too short response time, missing values, and the same answer to the main variable, 752 valid online questionnaires were obtained. The effective rate of the questionnaire was 91.2%. Finally, the 752 valid questionnaire data were collected for questionnaire analysis. The demographic data of the participants is shown in Table 1.

Respondents are gender balanced, with males accounting for 41.9% and females accounting for 58.1% (n = 435). The majority of the sample (n = 671, 75.1%) was between 18 and 35 years old, and most of the respondents had at least a university education (n = 426, 56.6%). Considering that these respondents are the main components of the Chinese MSVA fashion product

**Table 1. Consumer characteristics.**

| Demographic | Frequency | Percentage(%) | Demographic | Frequency | Percentage(%) |
|---|---|---|---|---|---|
| Gender | | | Education | | |
| Male | 315 | 41.9 | High school or below | 112 | 14.9 |
| Female | 437 | 58.1 | undergraduate | 426 | 56.6 |
| Age | | | Master or above | 214 | 28.5 |
| < 18 | 102 | 13.6 | Monthly income (RMB) | | |
| 18–25 | 204 | 27.1 | < 2000 | 54 | 7.2 |
| 26–30 | 236 | 31.4 | 2001–4000 | 178 | 23.7 |
| 31–35 | 125 | 16.6 | 4001–6000 | 146 | 23.4 |
| > 36 | 85 | 11.3 | 6001–8000 | 223 | 29.7 |
| Occupation | | | 8001–10000 | 102 | 13.4 |
| Student | 153 | 20.3 | > 10,000 | 49 | 6.5 |
| Manager or above | 108 | 14.4 | Purchase experience | | |
| Civil servant | 114 | 15.2 | Coat | 572 | 76.1 |
| Teacher | 96 | 84.2 | Dress | 465 | 61.8 |
| Clerk | 138 | 18.4 | Trousers | 527 | 70.1 |
| Blue-collar | 107 | 14.2 | Accessories | 618 | 82.2 |
| Others | 36 | 4.8 | Others | 104 | 13.6 |

market, this study believes that the samples are suitable for this study. In addition, the sample set includes a wide range of occupations (for example, students, managers, civil servants, teachers, clerks/white-collar workers, blue-collar workers), and their monthly income is mostly diversified.

## Measures

The scale of this study is taken from existing literature and fine-tuned according to the context. Media interactivity is measured by five items, which are adapted from Dong (2018) [57] and Abdullah et al. (2016) [49]. As media interactive is the critical single stimulus variable, we try to enrich the scale of media interaction. It involves the interaction between people and people, between people and machines, and between people and objects. Perceived value refers to consumers' overall assessment of the benefits and costs of energy-saving products. The measurement items of perceived value are adapted from Cuong (2020) [37] and Chae (2016) [38], and the scale design process of perceived value extends from perceived functional value to perceived entertainment value. The measurement items of immersion experiences are adapted from Charfi (2014) [52]. Purchase intention refers to the probability that consumers will try to buy fashion products under the stimulation of MSVA interaction. The measurement items of purchase intention are adapted from Chen (2018) [12] and Hajli (2019) [58]. A 5-point Likert scale was used for all measurement items, with 1 expressing strong disagreement and 5 expressing strong agreement.

## Results and discussion

Firstly, descriptive statistical analysis and exploratory analysis were carried out on all scale items to obtain the mean, correlation, and principal components of variables. Then, confirmatory factor analysis (CFA) was used to test the reliability and validity of the model. Finally, the structural equation model (SEM) was used to test the relationship between the proposed structures: media interactivity, perceived value, immersion experience, and purchase intention.

**Table 2. Descriptive statistics and reliability test of all items.**

| | Media Interactivity | Perceived Value | Immersion Experience | Purchase Intention |
|---|---|---|---|---|
| **Media Interactivity** | 0.724 | | | |
| **Perceived Value** | 0.542** | 0.757 | | |
| **Immersive Experience** | 0.573** | 0.575** | 0.763 | |
| **Purchase Intention** | 0.524** | 0.535** | 0.538** | 0.748 |
| **Means** | 3.571 | 3.316 | 3.865 | 3.384 |
| **SD** | 0.914 | 0.945 | 0.951 | 0.882 |

Note: The diagonal boldface is the square root of AVE

*shows significance at the 0.05 level

**shows significance at the 0.01 level.

## Reliability and validity test

The descriptive and reliability statistics of all items are shown in Table 2. According to the results of CFA, we obtained the overall fit index. These results indicated that the Chi-square/df was 2.284 and less than 3.00, GFI was 0.903, NFI was 0.915, TLI was 0.924, AGFI was 0.935, CFI was 0.912, and RMSEA was 0.04. All indices have reached the standard meaning that the measurement model was acceptable.

The construct reliability can be measured by Cronbach's alpha value and the composite reliability value. As described in Table 3, the composite reliability values ranged from 0.764 to 0.886, and the Cronbach's alpha values ranged from 0.791 to 0.869. The recommended values of the composite reliability and the Cronbach's alpha are 0.70, which indicates that the reliability of the variable is sufficient. Loadings are above the threshold level of 0.70, and the values of the AVE are larger than 0.50, which shows that the convergent validity of the scale is very well. Moreover, the square root of AVE of the individual variable was greater than the shared

**Table 3. Means, standard deviations (SD) and correlations.**

| Factors and Items | Loading | CR | Cronbach's α | AVE |
|---|---|---|---|---|
| **Media Interactivity** | | | | |
| I am a member of the MSVA and can communicate freely with others. | 0.837 | 0.871 | 0.869 | 0.727 |
| I can establish good social relations with other users and become. friends. | 0.864 | | | |
| There are many activities and services about product information. exchange and interpersonal interaction. | 0.886 | | | |
| There are a lot of interaction design features, which can help me. achieve the expected goals. | O.838 | | | |
| The MSVA can give me valuable recommendations and suggestions. whenever I need. | 0.815 | | | |
| **Perceived Value** | | | | |
| The clothing purchased in MSVA can offer great value. | 0.847 | 0.863 | 0.791 | 0.684 |
| The clothing purchased in MSVA can meet my expectations. | 0.826 | | | |
| The interactive experience of buying clothes in MSVA is very good. | 0.865 | | | |
| **Immersion Experience** | | | | |
| When I participated in interaction in MSVA, I felt that time passed quickly. | 0.764 | 0.836 | 0.835 | 0.705 |
| I really enjoy the process of buying clothing in MSVA. | 0.828 | | | |
| The MSVA has a strong attraction for me to buy clothing. | 0.852 | | | |
| **Purchase Intention** | | | | |
| I will always use the MSVA to buy clothing in the future. | 0.861 | 0.878 | 0.856 | 0.742 |
| I will use MSVA rather than other types of e-commerce to keep buying clothing. | 0.785 | | | |
| I would recommend MSVA to others to buy clothing. | 0.842 | | | |

**Table 4. Path coefficients of the structural model.**

| Item | Principal Components(PC) | | | |
|---|---|---|---|---|
| | **PC1** | **PC2** | **PC3** | **PC4** |
| Q4 | 0.713 | 0.131 | 0.176 | 0.225 |
| Q1 | 0.689 | 0.039 | 0.168 | 0.126 |
| Q2 | 0.674 | 0.028 | 0.181 | 0.138 |
| Q3 | 0.632 | 0.127 | 0.254 | 0.018 |
| Q5 | 0.611 | 0.064 | 0.035 | 0.142 |
| Q6 | 0.145 | 0.722 | 0.132 | 0.100 |
| Q7 | 0.198 | 0.685 | 0.066 | 0.031 |
| Q8 | 0.083 | 0.634 | 0.127 | 0.059 |
| Q10 | 0.139 | 0.125 | 0.793 | 0.151 |
| Q9 | 0.224 | 0.248 | 0.746 | 0.052 |
| Q11 | 0.018 | 0.138 | 0.603 | 0.162 |
| Q12 | 0.133 | 0.117 | 0.073 | 0.808 |
| Q14 | 0.167 | 0.142 | 0.249 | 0.784 |
| Q13 | 0.253 | 0.109 | 0.164 | 0.637 |
| **Eigenvalue** | 3.418 | 2.742 | 2.283 | 1.408 |
| **% of variance** | 29.413 | 23.642 | 18.774 | 13.562 |
| **cumulative % of variance** | 29.413 | 53.055 | 71.829 | 85.391 |

variances of the inter-construct and correlations between the variables, which supports the acceptability of discriminant validity (as showed in Table 3). These results indicated that the reliability and validity of each construct are acceptable.

## Exploratory analysis

In this study, exploratory factor analysis was conducted by non-rotating principal component analysis. The results showed in Table 4, and the explanatory variation of the first factor was 29.413%, which met the test criteria, and the factor load of independent variable and dependent variable did not appear on the same factor. At the same time, this paper also refered to the common method deviation test method adopted by Pavlou et al. (2007) [59]. If there is a common method deviation, the correlation of variables is greater than 0.9. The variable correlation matrix of this study (as showed in Table 3) showed that the correlation coefficients between variables are less than 0.9, indicating that there is no common method deviation. Based on the above analysis, the common method deviation of the formal survey sample data of this study is acceptable.

## Model hypothesis test

The result of model fitness analysis revealed that the Chi-square/df was 2.235 and the other indicators (GFI = 0.925, NFI = 0.914, IFI = 0.918, AGFI = 0.902, CFI = 0.927, and RMSEA = 0.05) were also acceptable. The path diagram of conceptual model is shown as Fig 2. It shows that the model of the study fits very well with the structural model. As shown in Table 5, Media Interactivity receptivity positively and significantly influences perceived value ($\beta$ = 0.363, t = 2.516, p<0.001), immersion experience ($\beta$ = 0.472, t = 2.162, p<0.001), and fashion purchase intention ($\beta$ = 0.254, t = 4.257, p<0.001). Thus H1, H2, and H3 are supported. Perceived value has a positive effect on immersion experience ($\beta$ = 0.372, t = 3.263, p<0.001) and purchase intention to buy fashion products ($\beta$ = 0.218, t = 2.327, p<0.001).

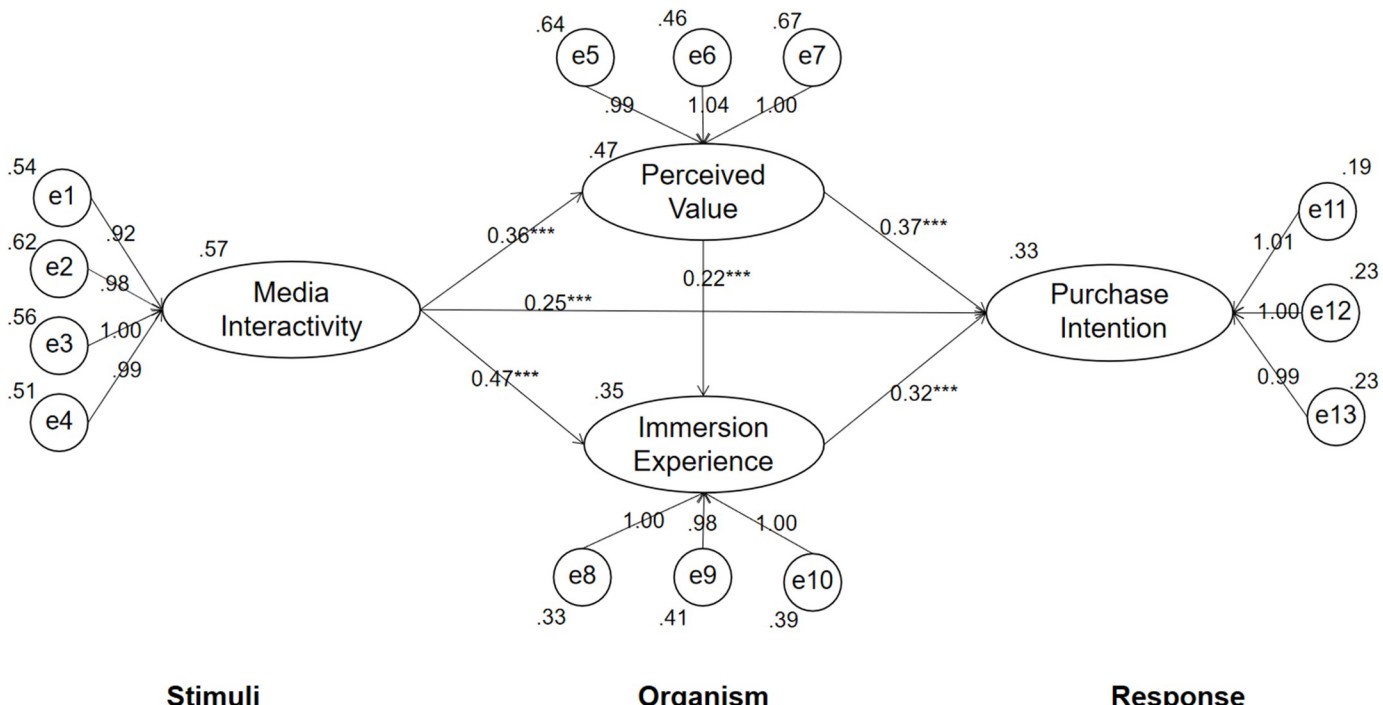

**Fig 2. Path diagram of conceptual model.**

These results support H4 and H5. Immersion experience has a positive effect on purchase intention ($\beta = 0.317$, t = 3.168, p<0.001) to buy clothing, thus, H6 is supported.

## Mediating effect analysis

As shown in Fig 2, to test the mediating effects of perceived value and immersion experience in this study, the bootstrapping analysis was done with AMOS 21.0. As described in Table 6 (the number of samples is 5,000 with a confidence level of 95%), the results did not cover 0, which indicated that social media interactivity has a significant indirect impact on continuous purchase intention through perceived value and immersive experience. Thus, the influence of media interactivity on consumers' purchase intention is mediated by perceived value and

**Table 5. Path coefficients of the structural model.**

| Hypothesis Path | β | S.E. | t-Value | P | Results |
|---|---|---|---|---|---|
| Media Interactivity→Perceived Value | 0.363 | 0.139 | 2.516 | *** | Supported |
| Media Interactivity→Immersion Experience | 0.472 | 0.126 | 2.162 | *** | Supported |
| Media Interactivity→Purchase Intention | 0.254 | 0.102 | 4.257 | *** | Supported |
| Perceived Value→Immersion Experience | 0.372 | 0.114 | 3.263 | *** | Supported |
| Perceived Value→Purchase Intention | 0.218 | 0.131 | 2.327 | *** | Supported |
| Immersion Experience→Purchase Intention | 0.317 | 0.164 | 3.168 | *** | Supported |

Note

* p < 0.05

** p < 0.01

*** p < 0.001.

Table 6. The mediating effect of perceived value.

| Mediation paths | Indirect effects | Lower bound | Upper bound | P-value |
|---|---|---|---|---|
| Media Interactivity→Perceived Value→Purchase Intention | 0.184 | 0.119 | 0.265 | *** |
| Media Interactivity→Immersion Experience→ Purchase Intention | 0.176 | 0.093 | 0.252 | *** |
| Perceived Value→Immersion Experience→Purchase Intention | 0.158 | 0.136 | 0.247 | *** |

Note

\* p < 0.05

\*\* p < 0.01

\*\*\* p < 0.001.

immersive experience. Perceived value has a significant indirect impact on purchase intention through immersive experience. Thus, the influence of perceived value on consumers' fashion purchase intention is mediated by immersion experience.

## Discussion

On the basis of the S-O-R theory, this research explored how MSVA media interactivity influences consumers' purchase intention. Specifically, this study not only revealed the "black box" between MSVA media interactivity and consumers' purchase intention through the study of perceived value and immersion experience, but also examined the boundary conditions of the impact of MSVA media interactivity on the intention to purchase fashion products on MSVA.

First, we found that media interactivity has a positive impact on perceived value, immersion experience and purchase intention. These results are consistent with previous research results (Abdullah et al., 2016 [49]; Huang and Liao 2017 [50]). When consumers browse the mobile short video application, they are easily attracted by the MSVA host, and will be curious about the purchase channels and intensive collocation of the clothes worn by the host. At this time, consumers expect answers through online questions or messages, and other users or MSVA host will reply to questions after seeing the questions, so as to realize or promote clothing consumption. In this process, consumers feel the color, material, and dressing effect of clothing through short video applications, and the learning cost is very low to improve the perceived value and immersion experience. Specifically, MSVA media interactivity can influence consumers' cognition of MSVA. Moreover, MSVA media interactivity highlights that fashion products are beneficial to perceived value, so they can stimulate consumers' desire to use MSVA and lead to consumers' immersion experience. This is because the reliability and attractiveness conveyed by MSVA media interactivity plays an important part in meeting the needs and wishes of consumers.

Second, the results indicated that perceived value and immersion experience have a significant positive impact on purchase intention. The results are consistent with the findings of Li et al. (2021) [29] and Zhou (2020) [56]. When consumers watch the video, it is easy to think of their state of wearing the clothes in the video. Generally, they think they are leisure and happy when watching the video, feel strong entertainment value, and save their time shopping and buying clothes. Consumers think they have gained more. On the contrary, if there is no impact of short video, they cannot perceive the value of fashion products and believe that the benefits they get are far less than the costs paid, so they are reluctant to buy this cloth. Immersion experience can play a significant part in buying behavior. The short videos released by the MSVA have time control, each video has a relatively complete plot, which has a strong attraction to consumers, so consumers often spend a lot of time watching short videos. Many times, consumers do not intend to buy cloth, but are inadvertently aroused when watching videos,

resulting in shopping behavior. Consumers affected by immersion experience believe that they can enjoy the process of buying. Therefore, immersion experience is positively correlated with consumers' purchase intention.

Third, perceived value has a positive impact on immersion experience. The result is consistent with the findings of Charfi (2014) [52]. When shopping online, consumers can determine whether they are suitable for themselves through the clothing information introduced by the platform, so as to decide whether to buy. When shopping with short video applications, consumers can more intuitively feel the various states of clothes after being worn, such as length, neckline size, pocket position, etc. On one hand, consumers feel higher practical value when buying clothes with short video applications. On the other hand, short video brings consumers a lot of entertainment value, so it is easy to have a strong immersion experience.

## Conclusion

This research expands the current knowledge on how to carry out clothing marketing on MSVA. More specifically, from the perspective of social system, it explores the factors that can promote consumers' clothing purchase intention. Using S-O-R model, we identified external cues as media interactivity. In addition, we also studied how media interactivity stimulates clothing purchase intention through consumers' perceived value and immersion experience. Finally, we emphasized the importance of media interaction to the success of e-commerce. Some theoretical and practical implications are also discussed.

### Implications

This study extends the current knowledge to the understanding of modern clothing e-commerce behavior. Firstly, this study enriches the research scenario of purchase intention by investigating consumers' clothing purchase intention on MSVA. MSVA has attracted hundreds of millions of users to create and consume short videos, but practitioners and researchers in this field pay more attention to the adoption of this form of social media and ignore their potential in clothing e-commerce. Therefore, this study expands the research on MSVA clothing e-commerce behavior by providing a conceptual model combining MSVA interactivity, perceived value, immersion experience, and purchase intention. The research model reflects the integration of MSVA's advantages in e-commerce and social media. In addition to previous e-commerce studies (such as Johnson, 2013 [20]; Nash, 2019 [21]; Changhan et al., 2021 [23]), this study emphasizes the importance of media interaction in influencing consumers' clothing purchase intention. In the previous research on clothing purchase intention, perceived value is usually combined with trust. However, this study adopts perceived value and immersion experience because immersion experience is more closely related to the media interaction of MSVA. Thus, the conceptual model is more suitable for the current e-commerce environment and lays a theoretical foundation for the future research on MSVA.

Based on this research, developers of mobile short video applications can develop a more user-friendly system. Meanwhile, marketing users of mobile short video applications can publish more efficient video works and enhance consumers' purchase intention. First of all, from the perspective of perceived value, consumers' perceived functional value and entertainment value can enhance their purchase intention, which means that mobile short video application designers and video creators need to provide consumers with an emotional shopping environment. From the perspective of immersive experience, the entertainment that consumers feel in mobile short video applications can enhance their immersive experience and enhance their purchase intention. Therefore, short video creators need to pay attention to the entertainment and plot of short video to avoid the direct whitening of advertising effect. From the perspective

of purchase intention, mobile short video applications are more vivid, interesting and relaxed than traditional e-commerce.

From the perspective of external stimulation, video creators should understand consumers. Video content should not involve too much superficial product information, but should stimulate consumers' sense of participation and encourage consumers to explore product details, so as to obtain perceived value and immersion experience. For system designers, it is necessary to increase high-quality and personalized video traffic, expand user labels, beautify the user interface and increase the user's immersion experience, which can increase consumers' willingness to buy.

## Limitations

The limitations of this study are, first, the influencing factors of purchasing intention in this paper are from the perspective of stimulus-organism-response. The influence of non-cognitive factors, such as personal habits and moral norms is not considered. Future research will be focused on non-cognitive factors. Second, this research was conducted in China. Further research will be done in other countries to determine whether the results of this research can be extended to the situations of other countries. Finally, the data of this study is cross-sectional data, which might not show the dynamic relationships among MSVA media interactivity, perceived value, immersion experience, and purchase intention in the social e-commerce field, and media interaction is the critical single stimulus variable. In the future, we will conduct more in-depth research on multiple stimulus variables, respondents from different countries and non-cognitive factors.

## Supporting information

**S1 Data.**
(XLSX)

## Author Contributions

**Investigation:** Tian Hewei.

**Writing – original draft:** Tian Hewei.

**Writing – review & editing:** Tian Hewei.

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
