## [Decision Letter · Decision Letter 0]

23 Nov 2021

PONE-D-21-31016Factors Affecting Continuous Purchase Intention of Fashion Products on Social E-commerce: Moderating Effect of Fashion InvolvementPLOS ONE

Dear Dr. Hewei,

Thank you for submitting your manuscript to PLOS ONE. After careful consideration, we feel that it has merit but does not fully meet PLOS ONE’s publication criteria as it currently stands. Therefore, we invite you to submit a revised version of the manuscript that addresses the points raised during the review process. Please follow reviews with special focus on the theoretical contribution in the current manuscript and managerial applications. Please submit your revised manuscript by Jan 07 2022 11:59PM. If you will need more time than this to complete your revisions, please reply to this message or contact the journal office at plosone@plos.org. Please include the following items when submitting your revised manuscript:A rebuttal letter that responds to each point raised by the academic editor and reviewer(s). You should upload this letter as a separate file labeled 'Response to Reviewers'.A marked-up copy of your manuscript that highlights changes made to the original version. You should upload this as a separate file labeled 'Revised Manuscript with Track Changes'.An unmarked version of your revised paper without tracked changes. You should upload this as a separate file labeled 'Manuscript'.

We look forward to receiving your revised manuscript.

Kind regards,

Jarosław Jankowski

Academic Editor

PLOS ONE

Journal Requirements:

- https://www.igi-global.com/gateway/chapter/281522

The text that needs to be addressed involves the introduction.

In your revision ensure you cite all your sources (including your own works), and quote or rephrase any duplicated text outside the methods section. Further consideration is dependent on these concerns being addressed.

Reviewers' comments:

Reviewer's Responses to Questions

**Comments to the Author**

1. Is the manuscript technically sound, and do the data support the conclusions?

Reviewer #1: Yes

Reviewer #2: Yes

2. Has the statistical analysis been performed appropriately and rigorously? 

Reviewer #1: Yes

Reviewer #2: Yes

3. Have the authors made all data underlying the findings in their manuscript fully available?

Reviewer #1: Yes

Reviewer #2: No

4. Is the manuscript presented in an intelligible fashion and written in standard English?

Reviewer #1: Yes

Reviewer #2: Yes

5. Review Comments to the Author

Reviewer #1: 1. Abstract must contain: (a) Originality of the research; (b) Research objective; (c) Method; (d)

Empirical result; (e) Practical implications

2. Provide background information and set the context about your study in the context of research area (country), Introduce the specific topic of your research and explain why it is important, Mention past attempts to solve the research problem or to answer the research question and Conclude the Introduction by mentioning the specific objectives of your research.

3. 3. Specify methods and procedures of your study very clearly, as part of this section you can include research design, variables covered, data analysis method etc.

4. Prove the results and find a linkage among results, research questions and hypothesis by using previous studies/literature review, whether your hypothesis accepted or rejected, matches with similar results and dissimilar. If dissimilar, mention what cause.

5. References-Please follow the appropriate referencing system/ of journal’s guidelines accordingly.

Best of Luck!

Reviewer #2: This paper explores the relationship between interactivity of social e-commence, perceived value, immersive experience and continuous purchase intention based on the S-O-R framework for fashion products. The authors claim that social e-commence interactivity has a positive effect on consumers’ perceived value, immersive experience and continuous purchase intention of fashion products, perceived value influences immersive experience and continuous purchase intention, and immersive experience positively influences continuous purchase intention. The study is clear and significant. However, I have some concerns about this paper:

1. I doubt the theoretical contribution in the current manuscript. Past researches have shown the relationship between social media interactivity, perceived value, immersive experience and purchase intention (e.g., social media and purchase intention, Chang Ya Ping and Dong 2016; perceived interactivity and value, Li et al. 2021; perceived value and continuance intention, Changlin et al. 2020). So, the present research’s contribution hangs almost wholly on the novelty of the variable, continuous purchase intention. Authors should elaborate more about the concept of continuous purchase intention, especially about its difference from purchase intention.

2. The managerial application of the current research is limited. We have known that the interactivity of social media has many positive effects, including immersive experience, perceived value and purchase intention. Companies want to sell more products and the increased purchase intention of consumers are beneficial. Therefore, managers can utilize social media and maintain its interactivity. But what is the difference between continuous purchase intention and purchase intention? I advise that authors pay more attention to the suggestions about how to increase the interactivity of social media and the significance of the continuity of purchase intention.

3. I do not think the elaboration of the theory development is specific enough. For H1, social media interactivity has a significant impact on continuous purchase intention, the theory supporting this hypothesis only relies on the research of Chang Ya Ping and Dong (2016), and Gasawneh et al. (2020) who showed the social media positively affect the purchase intention. How the interactivity or social media influences the continuous purchase intention is missing. Authors should explain in detail how the effect of social media is working, i.e., the underlying process. The same as other hypotheses, authors should explain the theory development in more detail.

4. The research focuses on fashion products. I am wondering how you defined fashion products, and whether it is different from other kinds of products. Since the consumers of the fashion industry tend to have more social needs, interactivity seems to have a greater impact on consumers. Maybe the authors can discuss this in the future research part.

5. The whole writing should be polished, as there are several grammar errors in the manuscript.

I therefore hope these comments, questions, and suggestions will be useful in helping you refine the manuscript.

Kind regards and best of luck!

6. PLOS authors have the option to publish the peer review history of their article (what does this mean?). If published, this will include your full peer review and any attached files.

Reviewer #1: No

Reviewer #2: No

---

## [Author Response · Author response to Decision Letter 0]

12 Apr 2022

Thank you for your patience and kind suggestions on our manuscript. We have revised the Manuscript PONE-D-21-31016 exactly according to the comments, and found these comments are very helpful. 

According to the suggestions, we have revised the manuscript according to the comments of the reviewers. We hope this revision can make our paper more acceptable. Special thanks to you for your careful comments. The revisions were addressed point by point below. 

Lists of Responses

Reviewer #1: 1. Abstract must contain: (a) Originality of the research; (b) Research objective; (c) Method; (d)

Empirical result; (e) Practical implications

Responses: Thanks for your careful revision and thanks a lot for your suggestions. The abstract has been reorganized and I hope this revision will contribute to the article.

2. Provide background information and set the context about your study in the context of research area (country), Introduce the specific topic of your research and explain why it is important, Mention past attempts to solve the research problem or to answer the research question and Conclude the Introduction by mentioning the specific objectives of your research.

Responses: Thanks for your careful revision and thanks a lot for your suggestions. We have rewritten the introduction of the paper, and it has been revised in the text.

3. 3. Specify methods and procedures of your study very clearly, as part of this section you can include research design, variables covered, data analysis method etc.

Responses: Thanks for your careful revision and thanks a lot for your suggestions. We have rewritten the methods and procedures of the paper, and it has been revised in the text.

4. Prove the results and find a linkage among results, research questions and hypothesis by using previous studies/literature review, whether your hypothesis accepted or rejected, matches with similar results and dissimilar. If dissimilar, mention what cause.

Responses: Thanks for your careful revision and thanks a lot for your suggestions. We have linked relevant studies, explain whether they are consistent or inconsistent with our research conclusions, and explain the relevant reasons, and it has been revised in the text.

5. References-Please follow the appropriate referencing system/ of journal’s guidelines accordingly.

Responses: Thanks for your careful revision and thanks a lot for your suggestions. We have revised and reorganized the references of the paper, and it has been revised in the text.

Reviewer #2: This paper explores the relationship between interactivity of social e-commence, perceived value, immersive experience and continuous purchase intention based on the S-O-R framework for fashion products. The authors claim that social e-commence interactivity has a positive effect on consumers’ perceived value, immersive experience and continuous purchase intention of fashion products, perceived value influences immersive experience and continuous purchase intention, and immersive experience positively influences continuous purchase intention. The study is clear and significant. However, I have some concerns about this paper:

1. I doubt the theoretical contribution in the current manuscript. Past researches have shown the relationship between social media interactivity, perceived value, immersive experience and purchase intention (e.g., social media and purchase intention, Chang Ya Ping and Dong 2016; perceived interactivity and value, Li et al. 2021; perceived value and continuance intention, Changlin et al. 2020). So, the present research’s contribution hangs almost wholly on the novelty of the variable, continuous purchase intention. Authors should elaborate more about the concept of continuous purchase intention, especially about its difference from purchase intention.

Responses: Thanks for your careful revision and thanks a lot for your suggestions. In the process of writing this paper, we did not highlight the contribution of this paper, which made this paper too dependent on previous research. In fact, we not only reorganize the variables, but also pay attention to the latest form of social e-commerce. This paper focuses on the clothing purchase intention of mobile short video application. Therefore, this paper has good theoretical value and practical significance for fashion marketing and mobile short video application marketing. Through the guidance of the reviewers, we can highlight the contribution of the paper and better translate the names of variables

2. The managerial application of the current research is limited. We have known that the interactivity of social media has many positive effects, including immersive experience, perceived value and purchase intention. Companies want to sell more products and the increased purchase intention of consumers are beneficial. Therefore, managers can utilize social media and maintain its interactivity. But what is the difference between continuous purchase intention and purchase intention? I advise that authors pay more attention to the suggestions about how to increase the interactivity of social media and the significance of the continuity of purchase intention.

Responses: Thanks for your careful revision and thanks a lot for your suggestions. We highlight the link between media interactivity and continued purchase intention, and it has been revised in the text.

3. I do not think the elaboration of the theory development is specific enough. For H1, social media interactivity has a significant impact on continuous purchase intention, the theory supporting this hypothesis only relies on the research of Chang Ya Ping and Dong (2016), and Gasawneh et al. (2020) who showed the social media positively affect the purchase intention. How the interactivity or social media influences the continuous purchase intention is missing. Authors should explain in detail how the effect of social media is working, i.e., the underlying process. The same as other hypotheses, authors should explain the theory development in more detail.

Responses: Thanks for your careful revision and thanks a lot for your suggestions. We feel that the hypotheses of the manuscript are not very clear, which brings a lot of trouble to the reviewer's review. After the guidance of the reviewer, we are impressed, so we rewrite all the hypotheses, and it has been revised in the text.

4. The research focuses on fashion products. I am wondering how you defined fashion products, and whether it is different from other kinds of products. Since the consumers of the fashion industry tend to have more social needs, interactivity seems to have a greater impact on consumers. Maybe the authors can discuss this in the future research part.

Responses: Thanks for your careful revision and thanks a lot for your suggestions. We apologize for our lax statement. Because our research belongs to the field of fashion marketing, we directly use fashion. In fact, it should be more accurate to use clothing, and it has been revised in the text.

5. The whole writing should be polished, as there are several grammar errors in the manuscript.

Responses: Thanks for your careful revision and thanks a lot for your suggestions. We have invited native English editors to revise the grammar and expression of this paper. I hope this revision can make the expression of this paper more accurate, and it has been revised in the text.

---

## [Decision Letter · Decision Letter 1]

1 Jun 2022

PONE-D-21-31016R1Factors Affecting Clothing Purchase Intention in Mobile Short Video App: Mediation of Perceived Value and Immersion ExperiencePLOS ONE

Dear Dr. Hewei,

Thank you for submitting your manuscript to PLOS ONE. After careful consideration, we feel that it has merit but does not fully meet PLOS ONE’s publication criteria as it currently stands. Therefore, we invite you to submit a revised version of the manuscript that addresses the points raised during the review process.

Please follow review and update the managerial implications and provide a figure with the mediation analysis..

We look forward to receiving your revised manuscript.

Kind regards,

Jarosław Jankowski

Academic Editor

PLOS ONE

Reviewers' comments:

Reviewer's Responses to Questions

**Comments to the Author**

1. If the authors have adequately addressed your comments raised in a previous round of review and you feel that this manuscript is now acceptable for publication, you may indicate that here to bypass the “Comments to the Author” section, enter your conflict of interest statement in the “Confidential to Editor” section, and submit your "Accept" recommendation.

Reviewer #1: All comments have been addressed

Reviewer #2: All comments have been addressed

2. Is the manuscript technically sound, and do the data support the conclusions?

Reviewer #1: Yes

Reviewer #2: Yes

3. Has the statistical analysis been performed appropriately and rigorously? 

Reviewer #1: Yes

Reviewer #2: Yes

4. Have the authors made all data underlying the findings in their manuscript fully available?

Reviewer #1: Yes

Reviewer #2: Yes

5. Is the manuscript presented in an intelligible fashion and written in standard English?

Reviewer #1: Yes

Reviewer #2: Yes

6. Review Comments to the Author

Reviewer #1: (No Response)

Reviewer #2: The manuscript has been improved in this version. There are only some minor issues:

1. The managerial implications of this paper could be further elaborated.

2. A figure with the mediation analysis might be provided.

7. PLOS authors have the option to publish the peer review history of their article (what does this mean?). If published, this will include your full peer review and any attached files.

Reviewer #1: **Yes: **Md. Abu Issa Gazi, PhD

Associate Professor, School of Management

Jiujiang University, China

Reviewer #2: No

---

## [Author Response · Author response to Decision Letter 1]

15 Jun 2022

Lists of Responses

1.The managerial implications of this paper could be further elaborated.

Responses: Thanks for your careful revision and thanks a lot for your suggestions. We have further elaborated the managerial implications of this paper, and it has been revised in the text.

2. A figure with the mediation analysis might be provided.

Responses: Thanks for your careful revision and thanks a lot for your suggestions. We have made a figure with the mediation analysis (figure 2), and it has been revised in the text.

---

## [Editor Report · Decision Letter 2]

19 Aug 2022

Factors Affecting Clothing Purchase Intention in Mobile Short Video App: Mediation of Perceived Value and Immersion Experience

PONE-D-21-31016R2

Dear Dr. Hewei,

We’re pleased to inform you that your manuscript has been judged scientifically suitable for publication and will be formally accepted for publication once it meets all outstanding technical requirements.

Kind regards,

Yann Benetreau, PhD

Division Editor

PLOS ONE

Additional Editor Comments (optional):

When you submit your final version, please ensure to address the following requests:

* please ensure that all authors provide an institutional email address,

* please ensure that you consistently refer to yourself as 'I' instead of 'we' (we noticed one mention of 'we' in the methods section), and

* PLOS ONE does not copy edit accepted manuscripts. Please proofread for typos and grammar.
---

## [Editor Report · Acceptance letter]

4 Sep 2022

PONE-D-21-31016R2 

Factors Affecting Clothing Purchase Intention in Mobile Short Video App: Mediation of Perceived Value and Immersion Experience 

Dear Dr. Hewei:

I'm pleased to inform you that your manuscript has been deemed suitable for publication in PLOS ONE. Congratulations! Your manuscript is now with our production department. 

Kind regards, 

on behalf of

Dr. Yann Benetreau 

Staff Editor

PLOS ONE